# Yttrium-90 Internal Radiation Therapy as Part of the Multimodality Treatment of Metastatic Colorectal Carcinoma

**Michael P. Del Rosario [1,†], Nadine Abi-Jaoudeh [2,†], May T. Cho [1], Zeljka Jutric [3] and Farshid Dayyani [1,*]**

1.  Division of Hematology and Oncology, Department of Medicine, University of California Irvine, Orange, CA 92868, USA; mpdelros@gmail.com (M.P.D.R.); mayc5@hs.uci.edu (M.T.C.)
2.  Division of Interventional Radiology, Department of Radiology, University of California Irvine, Orange, CA 92868, USA; nadine@hs.uci.edu
3.  Division of Hepatobiliary and Pancreas Surgery, Department of Surgery, University of California Irvine, Orange, CA 92868, USA; zjutric@hs.uci.edu
*   Correspondence: fdayyani@hs.uci.edu
†   These authors contributed equally to this work.

**Simple Summary:** Most patients with advanced colorectal cancer have metastases (i.e., tumor spread) to the liver, and in the majority of the cases, liver failure will be the cause of death. While chemotherapy and biologic therapies are effective in controlling colorectal tumor cells, resistance invariably occurs. Other treatment modalities are needed. Colorectal cancer cells are sensitive to radiation. One approach to control liver metastases has been the use of small radioactive resin Yttrium-90 (Y-90) beads, which are selectively delivered via the liver artery to the tumors. While studies have shown tumor shrinkage with the resin Y-90 treatment, the optimal timing and the role of the combination of Y-90 with systemic therapies are not well-defined. In the current review, we summarize the available data for resin Y-90 treatment in advanced colorectal cancer and propose a possible approach of integrating resin Y-90 in the multimodality treatment of colorectal cancers.

**Abstract:** About 70% of patients with metastatic colorectal carcinoma (mCRC) have liver metastases. Hepatic failure accounts for most mCRC-related deaths. Therefore, controlling liver metastases may improve outcomes. A data overview of liver-directed treatment using yttrium-90 selective internal radiation therapy (SIRT) is provided as part of a multimodality treatment. SIRT in mCRC is discussed, and the prognostic factors for patient selection are defined. Pooled analyses of three recent trials incorporating SIRT plus chemotherapy revealed subsets of patients with mCRC who might benefit from SIRT. A multidisciplinary treatment for most mCRC patients is proposed to achieve long-term survival in this cohort of patients.

**Keywords:** colorectal liver metastases; radioembolization; Y-90 resin

## 1. Introduction

Colorectal cancer (CRC) is the third-most common cancer diagnosis and the second-leading cause of cancer-related mortality in the United States [1]. There are an estimated 145,600 cases each year, with a mortality rate of about 51,020 deaths per year [2]. Only 22% of patients diagnosed with CRC present with distant metastatic disease [2]. However, more than 70% of patients will ultimately develop liver metastases [3]. The prognosis for metastatic colorectal cancer (mCRC) is dismal, with a 5-year survival rate of 11% [3].

The predominant location for patients with metastasis from colon cancer is the liver because of venous drainage directly from the gastrointestinal tract [4]. Liver failure is the most common cause of death among patients with mCRC and accounts for about two-thirds of mCRC-related mortality [5]. This is partly due to metastatic tumor burden in the liver that replaces normal liver parenchyma, leading to a functional compromise with subsequent metabolic dysfunction. Additionally, well-described toxicity from commonly

used cytotoxic agents such as irinotecan (hepatic steatosis) and oxaliplatin (sinusoidal obstruction) further contribute to worsening liver function in this patient population [6]. Therefore, identifying treatment strategies that effectively control liver metastases while protecting normal liver function as much as possible are crucial to improve the overall survival (OS) in this patient population.

The management of unresectable mCRC typically involves several lines of cytotoxic agents with or without biologics. Molecular testing to determine *RAS*, *RAF*, and microsatellite instability can guide prognosis and treatment selection in individual patients, which may help improve prognosis [3]. A detailed discussion of the systemic treatment options is not in the scope of this review but can be found in the National Comprehensive Cancer Network (NCCN) and European Society for Medical Oncology (ESMO) guidelines [7].

It is well-established that multidisciplinary care and a multimodality approach to patients with hepatic mCRC can improve their outcomes. For example, in a study by Lan et al. [8], the 3-year survival rate improved from 25.6% to 38.2% after the multidisciplinary team was established for mCRC. Liver-directed treatment options for hepatic mCRC include surgical resection as the preferred intervention with the best long-term survival data [9] and can be combined with the thermal ablation (radiofrequency or microwave ablation) of small tumors deep in the liver parenchyma [10]. However, the majority of patients have more advanced disease that is not amenable to curative intent resection. Furthermore, more than half the patients who undergo hepatic resection ultimately experience disease recurrence [8]. Thus, other liver-directed therapies have been used to treat metastases and include ablation, stereotactic body radiation therapy (SBRT), hepatic arterial infusion, chemoembolization, and selective internal radiation therapy (SIRT) with yttrium-90 (Y-90). Percutaneous ablative strategies are often used in patients with fewer than three tumors that are smaller than 3 cm and that are unresectable due to comorbidities and reduced liver function. SBRT, while associated with good local disease control, has limited indications, as it applies to patients with oligometastatic disease who would typically be treated with ablation or surgery [11]. Hepatic tumors have a predominant arterial blood supply (>80%), whereas the normal liver parenchyma has predominant portal vein blood supply [12]. Moreover, they are radiosensitive; therefore, SIRT is an ideal modality used to treat these tumors. Compared to SBRT, SIRT also allows the treatment of multiple (>3) and large tumors. Healthy tissue remains relatively unaffected [13,14]. Thus, NCCN guidelines recommend SIRT on a limited basis for patients with hepatic mCRC who have previously undergone surgery [3].

There is a growing body of data on the role of SIRT in mCRC. Currently, the use of SIR-Spheres® Y-90 resin microspheres (Sirtex Medical Limited, Leonards, NSW, Australia) is approved for treating liver-dominant mCRC. We will review and summarize the available clinical data on Y-90 resin microspheres and SIRT and propose a potential treatment algorithm to guide the approach to patients with liver-dominant mCRC.

We sought to use the available published literature for radioembolization in the treatment for mCRC as a basis to combine with our own clinical experience to formulate a treatment paradigm for mCRC with liver metastases. We searched PubMed using the term "yttrium 90, colo*" and limited the results to the English language and clinical trials only. Currently, only SIR-Spheres® Y-90 resin microspheres have US Food and Drug Administration approval for mCRC in the United States, so we only included published manuscripts using SIR-Spheres® Y-90 resin microspheres. The manuscripts were independently reviewed by the first and last authors. The relevant study characteristics and endpoints were extracted and summarized in a descriptive fashion, as shown in Table 1. No formal statistical comparison was made.

**Table 1.** Published studies in mCRC using SIR-Spheres® Y-90 resin microspheres.

| Author | Year | P/R | Study Phase (1–3) | Ra/SA | Comparator | Primary Outcome | PFS | OS | Comments |
|---|---|---|---|---|---|---|---|---|---|
| Kennedy A.S. et al. [15] | 2015 | R | n/a | SA | n/a | n/a | n/a | 2nd line (13 mo), 3rd line (9.0 mo), 4th line (8.1 mo) | |
| Sharma R.A. et al. [16] | 2007 | P | 1 | SA | n/a | Toxicity | 9.3 mo | n/a | Y-90 Combined with FOLFOX4 |
| Van Hazel G.A. et al. [17] | 2004 | P | 2 | Ra | Fluorouracil/leucovorin | RR, time to PD, toxicity | n/a | 29.4 vs. 12.8 mo; $p$ = 0.002 | Time to PD (18.6 vs. 3.6 mo; $p$ < 0.0005) |
| Gray B. et al. [18] | 2001 | P | 3 | Ra | HAC | Any increased patient benefit | 9.7 vs. 15.9 mo; $p$ = 0.001 | n/a | The 1-, 2-, 3-, and 5-year survival for patients receiving SIR-Spheres® was 72%, 39%, 17% and 3.5% compared to 68%, 29%, 6.5%, and 0% for HAC alone |
| Wasan H.S. et al. [19] | 2017 | P | 3 | Ra | Oxaliplatin-based chemotherapy | OS | 10.3 vs. 11 mo | 23.3 vs. 22.6 mo | ORR any site (1.78; 95% CI 1.37–2.31; $p$ < 0.001); ORR liver (1.78; 95% CI 1.37–2.31; $p$ < 0.001) |
| van Hazel G.A. et al. [20] | 2016 | P | 3 | Ra | mFOLFOX6 | PFS | 10.2 vs. 10.7 mo | n/a | ORR any site (68.1% vs. 76.4%; $p$ = 0.113); ORR liver (68.8% vs. 78.7%; $p$ = 0.042) |
| Narsinh K. et al. [21] | 2014 | P | 2 | SA | n/a | PFS | 10.2 mo | 17.6 mo | Historical control cohort; median PFS 2.5 mo |
| Dunfee B.L. et al. [22] | 2010 | R | n/a | SA | n/a | prognostic factors | n/a | n/a | Y-90 Combined with irinotecan |
| Hendlisz A. et al. [23] | 2010 | P | 3 | Ra | 5-fluorouracil infusion | TTLP | 2.1 vs. 4.5 mo; $p$ = 0.03 | 7.3 vs. 10.0 mo; $p$ = 0.80 | 2.1 vs. 5.5 mo (HR 0.38; 95% CI 0.20–0.72; $p$ = 0.003) |
| Seidensticker R. [24] | 2012 | R | n/a | n/a | BSC | OS | n/a | 8.3 vs. 3.5 mo; $p$ < 0.001 | |
| Cosimelli M. et al. [25] | 2010 | P | 2 | SA | n/a | ORR | 3.7 mo | 12.6 mo | ORR liver (24.0%; 95% CI 12.2–35.8%; $p$ = 0.05) |
| Cohen S.J. et al. [26] | 2014 | P | 1 | SA | n/a | Dose escalation | 6.4 mo | 8.1 mo | Y-90 Combined with capecitabine |
| Sofocleous C.T. et al. [27] | 2014 | P | 1 | SA | n/a | Safety | 2.0 mo | 14.9 mo | |
| Benson A.B., 3rd et al. [28] | 2013 | P | 2 | SA | n/a | PFS | 2.9 mo | 8.8 mo | |
| Gulec S.A. et al. [29] | 2013 | P | 2 | n/a | Chemotherapy only | Decreased TLG | n/a | n/a | In vivo study; decreased TLG: 54.91% ± 38.55% vs. 90.55% ± 19.75% ($p$ < 0.01) |
| Dunfee B.L. et al. [30] | 2010 | R | n/a | SA | n/a | Prognostic factors | n/a | n/a | |
| Mulcahy M.F. et al. [31] | 2009 | P | 1 | SA | n/a | Safety | n/a | 14.5 mo | |
| Mancini R. et al. [32] | 2006 | P | 2 | SA | n/a | Toxicity/efficacy | n/a | n/a | Short follow up did not allow survival analysis |

mCRC: metastatic colorectal cancer; Y-90: yttrium-90; P: prospective; R: retrospective; Ra: randomized study; SA: single-arm study; PFS: progression-free survival; OS: overall survival; n/a: not applicable; mo: months; RR: response rate; PD: progressive disease; HAC: hepatic artery chemotherapy; ORR: objective response rate; CI: confidence interval; mFOLFOX6: modified fluorouracil, leucovorin, and oxaliplatin; TTLP: time to liver progression; HR: hazard ratio; BSC: best supportive care; TLG: total lesion glycolysis.

The definition of "liver-dominant" disease is somewhat subjective and certainly has to be individualized within the context of the specific circumstances for each patient. However, the following definition is based on the SIRFLOX trial and could serve as a guide in patient selection: extrahepatic metastases in the lungs (<5 nodules of ≤1 cm in diameter or a single nodule of ≤1.7 cm in diameter) and/or lymph node involvement in a single anatomic area of <2 cm in diameter).

## 2. Clinical Data for SIRT in mCRC Based on the Previous Lines of Therapy

### 2.1. First-Line Setting

Several clinical trials have evaluated the safety and efficacy of SIRT in previously untreated hepatic mCRC. These trials included early phase 1 studies, as well as large prospective randomized phase 3 trials incorporating SIRT in the first-line treatment of mCRC.

Sharma et al. [16] established the safety of combining SIRT with the FOLFOX4 (5-fluorouracil, leucovorin, and oxaliplatin) regimen in a phase 1 study. They included 20 chemotherapy-naive patients with inoperable liver mCRC. The patients were treated with modified FOLFOX4 systemic chemotherapy and SIRT. During the first three cycles, the patients were given escalating doses from 30 to 85-mg/m$^2$ oxaliplatin followed by full FOLFOX4 doses for cycles 4–12. The reported median progression-free survival (PFS) was 9.3 months, whereas the median time to progression (TTP) was 14.2 months. In this subpopulation, a maximum-tolerated dose of oxaliplatin for the first three cycles was 60 mg/m$^2$, followed by full FOLFOX4 doses, mainly based on the observed rates of neutropenia (grade 3 or 4 in 12 out of 20 patients). Other notable adverse events included grade 3 abdominal pain (5 out of 20 patients) and one episode of transient grade 3 hepatotoxicity.

In a small phase 2 randomized trial by van Hazel et al. [17], a total of 21 patients with a history of untreated advanced CRC liver metastases were randomized to receive either a single SIRT administration plus fluorouracil/leucovorin ($n = 11$) or fluorouracil/leucovorin only ($n = 10$). The overall response rate was significantly greater in the combination therapy group ($p < 0.001$). Additionally, the time to disease progression was 18.6 months in the SIRT group vs. 3.6 months in the chemotherapy-alone group ($p < 0.001$). Finally, the authors reported a longer median survival among patients receiving SIRT plus chemotherapy vs. chemotherapy only at 29.4 months vs. 12.8 months, respectively ($p = 0.025$).

Gray et al. [18] performed a phase 3 randomized clinical trial in 74 patients with bilobar, nonresectable liver metastases. The investigators compared the addition of a single administration of SIRT to regional hepatic arterial chemotherapy (HAC), which consisted of a 12-day infusion of floxuridine repeated at monthly intervals vs. a chemotherapy regimen only. When measured by tumor areas, the overall response rate was significantly higher in the SIRT plus HAC group vs. the HAC-only group (44% vs. 18%, respectively; $p = 0.01$). Additionally, patients receiving the SIRT plus HAC approach experienced a significantly longer time to disease progression in the liver (15.9 vs. 9.7 months, respectively; $p = 0.001$). A greater proportion of patients in the SIRT group had higher survival rates at 2 years, and the investigators also observed a trend toward improved survival among SIRT-treated patients who survived beyond 15 months ($p = 0.06$).

### 2.2. FOXFIRE Combined Analysis

The findings of the phase 1 study by Sharma et al. [16] led to a series of global phase 3 randomized trials with similar designs that currently comprise the largest prospective data on SIRT in the first-line setting for hepatic mCRC. The FOXFIRE, SIRFLOX, and FOXFIRE-Global trials were multicenter randomized phase 3 trials across Europe, New Zealand, Asia, and the United States. All three trials evaluated the addition of SIRT with Y-90 resin microspheres to a standard chemotherapy regimen with modified fluorouracil, leucovorin, oxaliplatin (mFOLFOX6), and bevacizumab in eligible patients. The FOXFIRE combined analysis integrated the data from all trials [19], more than 1100 patients, and is the largest randomized analysis in interventional oncology that addresses the question of whether the improved local control of liver metastases affects survival in mCRC. The

primary endpoint was the OS, with important secondary endpoints including the overall PFS, progression specifically in the liver, hepatic resection rate, adverse events, and quality of life measures.

Eligible patients included those with mCRC with liver-only or liver-dominant metastases, patients eligible for systemic chemotherapy as a first-line treatment, a World Health Organization performance status 0 to 1, limited extrahepatic disease, and a life expectancy ≥3 months. Limited extrahepatic disease was defined as up to five lung nodules amenable to future definitive treatment and/or a single site of extrahepatic disease, which could include multiple lymph nodes in one lymph node region. The primary tumor was in situ in about 50% of the patients enrolled.

The treatment schedule for the control arm included mFOLFOX6 with or without bevacizumab, and the experimental arm included mFOLFOX6 with a reduced dose of oxaliplatin (60 mg/m$^2$) for the first three cycles, followed by a total of 12 cycles of standard dose mFOLFOX6. In the experimental arm, bevacizumab was added only after four to six cycles (depending on the trial), and SIRT was given with cycle 1 or 2. Only about 1% to 2% of the patients received cetuximab instead of bevacizumab.

Patients randomized to receive SIRT required a hepatic angiogram and liver-to-lung shunt study prior to the SIRT procedure to determine their suitability to receive this treatment and to map the arterial system accurately. The prescribed activity of Y-90 resin microspheres administered for the SIRT treatment arm was tailored to each patient-based body surface area, the percentage of tumor involvement, and the magnitude of liver-to-lung shunting.

Considering the goal of the study was to assess the impact of liver-directed therapy on the OS in mCRC, it is important to highlight some of the features of the combined analysis. About 35% of the patients in each arm had extrahepatic metastases. Among all patients assigned to SIRT, 8.5% actually did not receive SIRT because of clinical deterioration (33.3%), aberrant vascular anatomy/lung shunting (40.0%), or withdrawal of patient consent for SIRT (20.0%). Patients in the SIRT arm received fewer cycles of full-dose oxaliplatin (43.8% vs. 49.1%), and fewer of them also received bevacizumab (35.6% vs. 46.6%).

In terms of the OS, the median survival for the chemotherapy-alone and chemotherapy plus SIRT arms were 23.3 months and 22.6 months, respectively. There was no significant difference between the two groups in terms of the number of events, as demonstrated by the pooled hazard ratio (HR) of 1.04 (95% confidence interval (CI) 0.90–1.19; $p = 0.609$).

Additionally, the overall PFS was similar between the two groups with a pooled HR of 0.90 (95% CI 0.79–1.02). The median PFS was 10.3 months in the chemotherapy-alone group and 11.0 months in the chemotherapy plus SIRT group. However, when looking at the first progression within the liver, there was a significant benefit for SIRT, with a reduction of liver progression from 50% to 30%, and an HR of 0.51 ($p < 0.001$). Conversely, extrahepatic progression occurred more commonly in the SIRT-treated group, potentially caused by suboptimal systemic control due to lower initial oxaliplatin doses and lower rate of biologics. However, these findings confirmed the positive effects of SIRT for controlling liver tumors.

An objective response was observed in more patients in the chemotherapy plus SIRT group than in the chemotherapy-alone group in each of the individual trials and in the combined analysis (72.2% vs. 63%; pooled odds ratio (OR) 1.52; 95% CI 1.18–1.96; $p = 0.0012$). Specifically, the odds of achieving an objective response in the liver were higher in the chemotherapy plus SIRT group than in the chemotherapy-alone group (pooled OR 1.78; 95% CI 1.37–2.31; $p < 0.001$).

In terms of adverse events, of 1078 patients who received at least one study treatment dose in the intent-to-treat population, 755 (70%) had a grade 3 or worse adverse event; specifically, 375 (74%) out of 507 patients in the chemotherapy plus SIRT group and 380 (67%) out of 571 patients in the chemotherapy-alone group. Adverse events occurred up to 28 days after the end of protocol chemotherapy or in the first 7 months after randomization (whichever was earlier). The odds of a patient having a grade 3 or worse adverse

event were higher in the chemotherapy plus SIRT group than in the chemotherapy-alone group (pooled OR 1.42; 95% CI 1.09–1.85; *p* = 0.0089). Out of 507 patients in the SIRT group, 231 (46%) had a hematological grade 3 or worse adverse event vs. 165 (29%) of the 571 patients in the chemotherapy-alone group. The most frequent adverse event was neutropenia, occurring in 186 (37%) patients in the chemotherapy plus SIRT group vs. 138 (24%) in the chemotherapy-alone group. The increased toxicity in the SIRT-treated patients did not translate into inferior OS. This study provides the most comprehensive account of the risk of adverse events that might occur secondary to SIRT when used in combination with mFOLFOX6 chemotherapy.

Recent studies have confirmed that tumor sidedness is prognostic and predictive for survival in mCRC, with a right-sided primary tumor location showing significantly worse survival outcomes than the left [33]. Preliminary data on the primary tumor location were prospectively gathered and reported as post hoc analysis from 719 out of 1103 patients in the FOXFIRE studies, comprising those in the SIRFLOX and FOXFIRE-Global cohorts [34]. The OS for patients with mCRC with right-sided primary tumors was significantly improved with the addition of SIRT to first-line mFOLFOX6 chemotherapy (median OS 22.0 vs. 17.1 months with or without SIRT, respectively; HR 0.64; 95% CI 0.46–0.89; *p* = 0.007).

In a previous study, the objective response to preoperative chemotherapy was found to correlate with the secondary resection rate [35]. In the SIRFLOX trial, the liver resection rate was not significantly different among both arms (13.7% chemotherapy vs. 14.2% chemotherapy plus SIRT; *p* = 0.857) [20]. However, based on the observed higher hepatic response rates with SIRT in the combined trials, a retrospective post hoc analysis was performed to independently establish resectability based on a surgeon-blinded review (REsect study) [36].

The REsect study was a blinded, retrospective imaging analysis done by surgeons' assessments of baseline and posttherapy resectability, including whether ablation combined with surgery would enable the removal of all liver metastases. The population consisted of 530 patients, with 472 evaluable and randomized to therapy. At the baseline, there were no differences in the rates of resectability between the chemotherapy vs. SIRT arm (11.0% vs. 11.9%; *p* = 0.775). However, after treatment, when evaluated in a blinded fashion by at least three hepatobiliary surgeons, SIRT plus chemotherapy was associated with a 9.2% greater technical resectability vs. chemotherapy only (28.9% vs. 38.1%; *p* < 0.001). Additionally, a secondary analysis of resectability was performed during the FOXFIRE cohort. The study described that microspheres were located in the periphery of the tumor and that patients with combination SIRT and chemotherapy had higher response rates and tumor necrosis as opposed to chemotherapy alone [37]. The study concluded that resection after SIRT is feasible.

### 2.3. Second-Line Setting

The MORE (Metastatic colorectal cancer liver metastases Outcomes after RadioEmbolization) study was an investigator-initiated study that retrospectively reviewed medical records for the safety and efficacy of Y-90 resin microspheres in 606 consecutive patients with mCRC at selected experienced centers in the United States between 2002 and 2015 [15]. The majority of the patients had a good performance status (Eastern Cooperative Oncology Group < 2): about one-third (35.1%) had extrahepatic metastases, and about 30% had more than 50% liver involvement by tumor volume [15]. The patients had up to five SIRT procedures. Importantly, patients at various stages of their treatment were included. Patients receiving second-line therapy constituted one-third of the cohort, another 30% were at their third-line treatment, and the final one-third were fourth-line and beyond. The median survival positively correlated with the line of treatment; earlier treatment with SIRT was associated with improved survival. The median survival in the second line was 13.0 months (95% CI 10.5–14.6 months) vs. 9.0 months for the third line (95% CI 7.8–11.0 months) and 8.1 months for the fourth line or greater (95% CI 6.4–9.3 months).

In the subset of 206 patients receiving second-line therapy in the MORE study, the survival following SIRT was 13.0 months, which is similar to the reported outcomes in phase 3 trials examining the chemotherapy plus biologic treatment of mCRC with second-line aflibercept (median 13.5 months) and bevacizumab beyond progression (median 11.2 months).

The MORE study identified important the prognostic factors for SIRT. A better performance status, absence of extrahepatic disease, earlier line of treatment, tumor burden less than 25% of the liver volume, increasing number of SIRT procedures, lack of ascites, and primary tumor were all independent prognostic factors of improved survival in this cohort. Interestingly, age was not prognostic, with a similar median survival of 9.7 months vs. 9.3 months for patients younger than 70 years compared to those older, without increased toxicity in the older population.

The most common adverse events were grades 1 and 2 and included abdominal pain (6.1%), followed by fatigue (5.5%), hyperbilirubinemia (5.1%), ascites (2.8%), vomiting (1.5%), and nausea (1.3%). This study also established the rate of serious adverse events (grade 4 or higher) for patients treated with SIRT. The most common were abdominal pain (6.1%), fatigue (5.5%), hyperbilirubinemia (5.1%), and ascites (2.8%), whereas the rates of other serious adverse events were less than 2% for each (including a 0.3% rate of liver failure).

The risks factors associated with the development of radioembolization-induced liver disease include whole-liver treatment, previous chemotherapy treatment, and excessive tumor burden [38]. The activity was calculated by the body surface area (BSA) method, which empirically accounts for liver and tumor volumes. This method is simple and is the recommended method by the manufacturer, but it tends to have shortcoming in small patients with big livers and big patients with small (i.e., cirrhotic) livers. Gil-Alzugaray et al. [38] recommended use of BSA when the entire liver will be treated with radiation and use of partition if two or more segments will be spared. The partition model assumes nonuniform distribution based on 99-m technetium macroaggregated albumin. The authors also reduced radioembolization-induced liver disease significantly by reducing the activity by 10% compared to BSA and significantly more if there was cirrhosis, previous chemotherapy, small tumor burden (<5%), or reduced liver volume (<1.5 L). If using the partition model with adequate remaining segments and liver function, the authors recommended a target dose of 100 Gy or greater in the treated area, whereas they advocated for a target dose of 40 Gy if a large area was treated or if the liver function was compromised (i.e., cirrhosis).

In an abstract presented at the 2014 annual scientific meeting of the Society of Interventional Radiology, Narsinh et al. [21] reported on whether administering Y-90 resin microspheres between first- and second-line irinotecan-based chemotherapy was associated with an increased time to PFS and higher tumor response rate when compared with second-line therapy among patients with liver-dominant mCRC. The median time to disease progression was 10.2 months, which was substantially greater than the median PFS of 2.5 months in the historical cohort, and the median OS was 17.6 months.

Another study by van Hazel et al. [22] sought to determine the maximum-tolerated dose of irinotecan plus SIRT among patients with CRC liver metastases who failed to respond to fluorouracil. For the first two cycles, irinotecan was provided at 50, 75, or 100 mg/m$^2$ on days 1 and 8 of a 3-week cycle. During cycles 3–9, the full 100-mg/m$^2$ irinotecan doses were administered. The investigators administered Y-90 resin microspheres during the first cycle of chemotherapy. Out of 25 patients, a total of three, five, and four patients receiving irinotecan 50, 75, and 100-mg/m$^2$ doses, respectively, experienced grade 3 or 4 adverse events. Approximately half of the patients (48%) had a partial response to therapy. The median PFS and median survival were 6.0 months and 12.2 months, respectively. Overall, the investigators were unable to reach a maximum-tolerated dose, recommending the 100-mg/m$^2$ irinotecan dose on days 1 and 8 during a 3-week cycle.

### 2.4. Chemorefractory Setting

In a prospective, phase 3 study by Hendlisz et al. [23], participants with chemorefractory, liver-limited mCRC were randomized to receive either an intravenous infusion of protracted 300-mg/m$^2$ fluorouracil for days 1–14 every 3 weeks ($n$ = 23) or SIRT plus 225-mg/m$^2$ intravenous fluorouracil for days 1–14 and then 300-mg/m$^2$ fluorouracil for days 1–14 every 3 weeks ($n$ = 21) until progression [23]. The median time to liver progression was significantly longer among those treated with SIRT vs. fluorouracil only (5.5 vs. 2.1 months, respectively; $p$ = 0.003), and the median time to tumor progression was also longer in the SIRT group (4.5 vs. 2.1 months; $p$ = 0.03). No significant difference was found between the two groups in terms of the median OS (10.0 vs. 7.3 months; $p$ = 0.80).

Another study found that SIRT plus the best supportive care ($n$ = 29) provides a significantly longer survival than the best supportive care only ($n$ = 29) [24]. The study's population included patients with chemorefractory, liver-dominant mCRC whose survival was significantly longer with SIRT and the best supportive care vs. the best supportive care only (8.3 vs. 3.5 months, respectively; $p$ < 0.001).

A multicenter, phase 2 clinical trial by Cosimelli et al. [25] evaluated Y-90 resin microspheres as a sole approach to unresectable, chemotherapy-refractory CRC liver metastases. Out of 48 patients, the overall response rate was 24.0%, as analyzed by the Response Evaluation Criteria In Solid Tumors ($p$ = 0.05). Additionally, the median TTP and PFS were approximately 4 months, whereas the median OS was 12.6 months. The 1-year survival rate was higher than the 2-year survival rate (50.4% vs. 19.6%, respectively). Finally, the median survival from the first CRC liver metastases diagnosis to death or the end of the study was 31 months. The lack of a comparator arm limited the findings of this study.

While small studies with limited patient numbers are prone to selection biases that might affect the observed survival and toxicity rates, it is noteworthy that an objective tumor response is the clinical endpoint most directly associated with this therapeutic intervention and least affected by other factors occurring outside of the treatment phase, such as subsequent treatments. In this regard, the clinical efficacy reported above for SIRT in the salvage setting compares favorably with published and approved oral systemic agents for mCRC, including regorafenib (objective response rate (ORR) 1%; median survival 6.4 months) and trifluridine/tipiracil (ORR 1.6%; median survival 7.1 months) [39,40]. Additionally, given more stringent inclusion criteria for these phase 3 registration trials, it is unlikely that patient selection for more "fit" patients accounts for the favorable clinical outcomes seen in the SIRT trials.

## 3. Conceptualizing a Roadmap to Long-Term Survival in mCRC

As with all the treatment options, appropriate patient selection is critical for the success of SIRT. As seen above, the ideal patients for SIRT appear to be those with non-resectable liver-only or liver-dominant disease, right-sided mCRC, a good performance status, adequate liver function, and expected survival greater than 3 months in the absence of any intervention. In addition, SIRT should not be offered during the initial cycles of chemotherapy but early in the treatment regimen for an added benefit to chemotherapy, biologic therapy, and other liver-directed therapy options.

The data reviewed previously may prompt further investigations for the optimization of multidisciplinary care of patients with hepatic mCRC, with the goal of improving their long-term survival. The expected rate of long-term survivors among all patients with mCRC is around 10–15%. Hence, a clinically meaningful improvement of this "survival curve tail" from 10% or 15% to potentially 20% to 25% would not necessarily improve the median OS in all comers. Thus, in the authors' opinions, the median OS does not appear to be the appropriate endpoint when evaluating a multidisciplinary approach to increase the proportion of long-term survivors.

Several paradigms were derived from clinical observations and/or the data discussed above and warrant further study of the treatment approach to patients with mCRC. The removal of all visible disease appears to be required for long-term survival [41,42]. Y-90

resin microspheres may possibly aid in achieving this goal. Further, in mCRC, the highest response rates to systemic treatment are seen in the first-line setting and are most pronounced in the first 3 months. After 3 months, further tumor shrinkage is unlikely, and, over time, a resistance develops, which leads to tumor growth in the liver and elsewhere. This implies that systemic therapy alone is unlikely to induce a durable complete response without other means of consolidation. On the other hand, SIRT is very effective in achieving a tumor response in the liver. Patients with a better performance status, lower disease volume in the liver, and earlier line of treatment have the best outcomes with SIRT. Therefore, SIRT appears to be most effective at about 3 months after systemic treatment for optimizing the highest chance of removal of the visible disease. This interesting hypothesis needs to be studied and evaluated in a prospective clinical trial. Indeed, the secondary analysis from FOXFIRE demonstrated that the combination group had less viable tumors than the chemotherapy-alone group.

From the above, the maximal systemic control with a doublet or triplet (e.g., mFOLFOX or FOLFOXIRI (i.e., fluorouracil, leucovorin, oxaliplatin, and irinotecan)) chemotherapy regimen plus a biologic (anti-EGFR or anti-VEGF, depending on the biomarker status and tumor-sidedness) for the first 3 months will produce the highest rates of disease response and control. After 3 months, the response curves plateau markedly, and consolidation of the disease sites becomes more important. Withdrawing oxaliplatin and/or irinotecan after the initial 3 months of chemotherapy should also avoid both resistance and excessive toxicity (bone marrow and neuropathy) due to prolonged exposure. NCCN and ESMO recommend discontinuation or periods of maintenance chemotherapy only for CRC patients.

For patients with liver-only or liver-dominant disease, especially with a very good response to systemic therapy and normalization of tumor markers, presentation at the multidisciplinary tumor board guides triage to liver resection with or without radiofrequency ablation vs. SIRT for unresectable patients. After control of the liver disease, the biology of the cancer can be tested by 2 to 3 months of additional maintenance chemotherapy (with or without a biologic), and if no progression, the patients can be referred for colorectal surgery to remove the primary tumor. Reimaging at this stage should demonstrate if SIRT produced resectable disease in the liver, which would lead to referral for hepatobiliary surgery. Ideally, a prospective clinical trial to test this hypothesis using an appropriate primary endpoint (i.e., 3- or 5-year survival rate rather than median survival) should be conducted.

### 4. Conclusions

SIRT can be a valuable addition to the multidisciplinary management of mCRC. Careful patient selection (i.e., performance status, disease volume, extrahepatic disease, and preserved liver functioan) and timing of the procedure may help maximizing the clinical outcomes.

**Author Contributions:** Conceptualization, methodology, and data curation: F.D. and M.P.D.R. Writing, review and editing: M.P.D.R., N.A.-J., M.T.C., Z.J., F.D. All authors have read and agreed to the published version of the manuscript.

**Funding:** Copyediting of this manuscript, prior to submission, was provided by Eubio through the support of Sirtex Medical, Inc. No Funding provided for this manuscript.

**Institutional Review Board Statement:** Not applicable.

**Informed Consent Statement:** Not applicable.

**Data Availability Statement:** Not applicable.

**Conflicts of Interest:** Del Rosario reports no conflict of interest. Abi-Jaoudeh reports part ownership in Bruin Biosciences and has received institutional research support from Philips, Teclison Cheery Pharma, and Sillajen. Cho has received institutional research support from BMS, Astrazeneca, and Incyte; served on the Speaker's Bureau for Taiho; and served on the advisory boards for Amgen, Taiho, Astellas, and Exelixis. Jutric reports no conflicts of interest. Dayyani served on the advisory

boards for Eisai, Genentech, Array, Exelixis, and Foundation Medicine; has received institutional research support from Taiho, AZD, Merck, BMS, and Exelixis; and served on the Speaker's Bureau for Amgen, Genentech, Sirtex, Ipsen, Eisai, and Exelixis.

## Abbreviations

BSA: body surface area; CI: confidence interval; CRC: colorectal cancer; ESMO: European Society for Medical Oncology; FOLFOX4: 5-fluorouracil, leucovorin, oxaliplatin; HAC: hepatic arterial chemotherapy; HR: hazard ratio; mCRC: metastatic colorectal cancer; mFOLFOX6: modified fluorouracil, leucovorin, and oxaliplatin; NCCN: National Comprehensive Cancer Network; OR: odds ratio; ORR: objective response rate; OS: overall survival; PFS: progression-free survival; SBRT: stereotactic body radiation therapy; SIRT: selective internal radiation therapy; TTP: time to progression; Y-90: yttrium-90.

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
