# Peer review of "Yttrium-90 Internal Radiation Therapy as Part of the Multimodality Treatment of Metastatic Colorectal Carcinoma"

_onco, doi:10.3390/onco1020015_

Round 1
Reviewer 1 Report
Thank you for the great work on a thorough review about Y-90 internal radiation therapy in the multimodality approach of the metastatic colon cancer. The review is structured in a well organized manner for the readers with introduction, reviewing the available trials based on the treatment line setting and then the conclusion.
--Can we include in the table for the single arm trial if they are combined with chemotherapy or SIRT as sole intervention. For example. Sharma et al, 2007 (second one in the table) SIRT was combined with chemotherapy?
Author Response
Thank you for your suggestion. For trials that combined Y-90 with chemotherapy, the regimen was added in Table 1 under the 'Comments' column
Reviewer 2 Report
This is an extensive review of the Yttrium-90 selective internal radiation therapy (SIRT) of liver metastases from colorectal cancer. The paper is well presented and very informative. There is a comprehensive analysis of studies on the subject. One particular significant point is the fact that only studies using the only FDA-approved Yttrium-90 microspheres were selected and analyzed. This greatly adds to the homogeneity and direct comparisons of reported results. All clinical aspects of SIRT have been clearly presented and the text is both succinct and detailed. There are no flaws in the use of English language and the conclusions are well documented. References are up to-date.
In summary, this is a well written and informative paper that will be very useful for all those involved in the management of patients with colorectal cancer.
Author Response
Thank you for your positive evaluation. No comments to address.
Reviewer 3 Report
Your paper reports a general review of this subject, without personal data. The authors are requested to reportand discuu their persomal experience.
Author Response
Thank you for your comment. The authors summarized their institutional approach in Section 3 of the manuscript (page 8, lines 334-385). A prospective collection of data for treated patients is ongoing and will be reported later.
We thank you very much for your consideration.

Round 2
Reviewer 3 Report
The new version is ameliorated